# Inner southern magnetosphere observation of Mercury via SERENA ion sensors in BepiColombo mission

Mercury's southern inner magnetosphere is an unexplored region as it was not observed by earlier space missions. In October 2021, BepiColombo mission has passed through this region during its first Mercury flyby. Here, we describe the observations of SERENA ion sensors nearby and inside Mercury's magnetosphere. An intermittent high-energy signal, possibly due to an interplanetary magnetic flux rope, has been observed downstream Mercury, together with low energy solar wind. Low energy ions, possibly due to satellite outgassing, were detected outside the magnetosphere. The dayside magnetopause and bow-shock crossing were much closer to the planet than expected, signature of a highly eroded magnetosphere. Different ion populations have been observed inside the magnetosphere, like low latitude boundary layer at magnetopause inbound and partial ring current at dawn close to the planet. These observations are important for understanding the weak magnetosphere behavior so close to the Sun, revealing details never reached before.

Planet Mercury was visited in the past by only two satellites: Mariner-10 (3 flybys in 1974 / 1975)[1], and 'MErcury Surface, Space ENvironment, GEochemistry, and Ranging' (MESSENGER), which orbited the planet from 2011 to 2015[2]. Concerning environment, Mariner-10 discovered the existence of a weak internal dipolar magnetic field;[3] MESSENGER allowed to quantify the magnetic dipole moment (190 nT $R_M$)[3], offset northward by about 0.2 $R_M$[4], and to depict a dynamic magnetosphere, strongly coupled with the solar wind conditions, and a high reconnection rate[5]. Anyway, none of the previous missions was able to fully explain the planet and environment peculiarities, so that many questions are still unsolved. The ESA-JAXA BepiColombo (BC) mission was launched in October 2018, having onboard a large set of instruments to better study the characteristics of this planet, so close to the Sun[6]. BC is composed by two elements: MPO (Mercury Planetary Orbiter, ESA), and Mio (Mercury Magnetospheric Orbiter, JAXA). After traveling in the interplanetary space for the first three years, BC passed by its target planet Mercury for the first time on 1st October 2021. The final orbital insertion of the two elements MPO and Mio will take place at the end of 2025: MPO will be inserted in a polar orbital path, at beginning between 480 and 1500 km; Mio will have a polar orbital path as well, at beginning between 590 and 11640 km. Before the beginning of the nominal phase, it will perform six Mercury Flybys in total[7]. In the actual cruise configuration of the composite spacecraft, not all BC instruments can operate. In particular, the 'Search for Exospheric Refilling and Emitted Natural Abundances' (SERENA) suite of four units, devoted to the study of the ion and neutral particle populations around the planet[8], has the possibility to perform scientific measurements during cruise via two units, 'Planetary Ion CAMera' (PICAM) and 'Miniaturized Ion Precipitation Analyzer' (MIPA), both devoted to the observation of positive ions coming from the solar wind as well as from the planet's environment. PICAM and MIPA have a 3D Field-of-View (FoV) < 2π, with the boresight pointing perpendicular to the Sun direction (see Supplementary Information for details). Both sensors are nominally able to detect the solar wind in their extreme lateral views: in this case, due to the sensitivity trend versus angle from the boresight, only PICAM is able to clearly detect the solar wind signal. Moreover, the two sensors together observe plasma regimes over a wide energy range, covering both solar wind and planetary ion populations, outside and inside Mercury's magnetosphere. Short technical feature descriptions of PICAM (Supplementary Fig. 1, Supplementary Table 1), and MIPA (Supplementary Fig. 2, Supplementary Table 2) are given in the Supplementary Information.

✉ e-mail: stefano.orsini@inaf.it

In the following, the timing of the observations along the BC trajectory near-by Mercury is described, and the PICAM and MIPA data are shown.

Here we show that the trajectory of the first Mercury flyby (MFB1) covers regions in the southern hemisphere at low altitudes not explored by previous missions. The collected data allow showing ion energy distributions at the bow shock and closer to Mercury in the southern hemisphere. Such preliminary raw data reveal very interesting solar wind features and magnetospheric plasma regimes, giving a clear evidence of the potentiality of BC instrumentation. MFB1 is a first relevant step versus a comprehensive understanding of the environment around Mercury.

## Results

### 1. BepiColombo trajectory and region traversals

The BC MFB1 occurred between the 1st and the 2nd of October 2021. The Mercury Solar Magnetospheric coordinated system (MSM) is centered on the planetary magnetic dipole with the *X*-axis positive in the solar direction and an offset northward along the MSM *Z*-axis by 480 km (about 0.2 RM), parallel to the planetary rotation axis[5]. The *Y*-axis is positive opposite to the direction of Mercury's orbital velocity which completes the right-handed MSM system. The spacecraft approached the planet from the dusk flank, the magnetosheath and near magnetotail, and exited the magnetosphere in the dawn dayside, again crossing the magnetosheath (Fig. 1). The closest approach occurred on October 1st, at 23:34 UT at an altitude of 199 km and $Z_{MSM}$ about −0.7 $R_M$ in the nightside. As shown in Fig. 2, PICAM operated during 4 distinct time periods and observed the solar wind ion flux (Panel a, insets 1 and 4), the inbound magnetosheath, and the inner magnetosphere (Panel a, insert 2), and the region upstream of the bow shock (Panel a, inset 3), while MIPA operated continuously from 22:35 UT to 23:56 UT, and observed the magnetosheath adjacent to the tail, the inner magnetosphere and the outbound magnetopause and bow shock (Panel b).

### 2. Solar wind observations

The solar wind was not always visible to PICAM and MIPA during the cruise, depending on the FoV direction (the FoV edge being about 30°

off the Sun direction). Nevertheless, while approaching Mercury, PICAM was able to see part of the solar wind distribution that appeared to be quite warm, dense, and at low energy (peaking at about 600 eV). Between 19:00 UT and 21:00 UT, at a distance of about 25 $R_M$ from Mercury center, in the dusk side, the spacecraft rotated and the PICAM boresight moved from the $-Z_{MSM}$ direction, i.e., the southern hemisphere to $+Z_{MSM}$ in the northern hemisphere (see Fig. 3). In doing so, PICAM FoV passed through the $-Y_{MSM}$ direction (i.e., moving to the same direction as the planet moves pointing along the ecliptic plane toward the bow shock).

During this time-period, PICAM observed clear intermittent features (with a time scale of a few minutes) at high energies (above 1 keV, Fig. 4a). Actually, their appearance is clearly associated with PICAM's FoV pointing towards the bow shock, as opposed to the solar wind direction, but the possibility that these intermittent structures could be related to a source from the bow-shock[9] is hardly applicable by considering that the vantage point is too far away from the bow shock itself. A combined analysis with magnetic field data from BC/MAG (MPO magnetometer) would be needed, to verify that these keV particles could be associated with the passage of an interplanetary magnetic flux rope with its axis oriented along the *Y*-axis. In this case, MAG should observe the typical signature of this structure, i.e., an increase of the average magnetic field magnitude (with respect to the main background field), a decrease of the variance of magnetic field fluctuations, and a smooth rotation of one of the field components. Such findings have a chance to be also validated by means of Solar Orbiter (SolO) magnetic field observations. In fact, SolO[10] was located at a distance of 0.64 AU from the Sun (0.26 AU ahead BC) and the two spacecraft were reasonably radially aligned, longitudinally separated by less than 10°, and lying on the same side of the heliospheric current sheet. Details of the results of this analysis will be reported in a forthcoming paper, as soon as the MAG data will be confirmed and officially validated. The actual effect over the Mercury environment would have been the subject of an interesting study, but unfortunately the solar wind structure vanished well before the flyby, and any possible internal effect was not observed. It likely produces enhanced flux transfer events and magnetic reconnection sites, together with small substorm-like activity in the nightside of the Hermean magnetosphere.

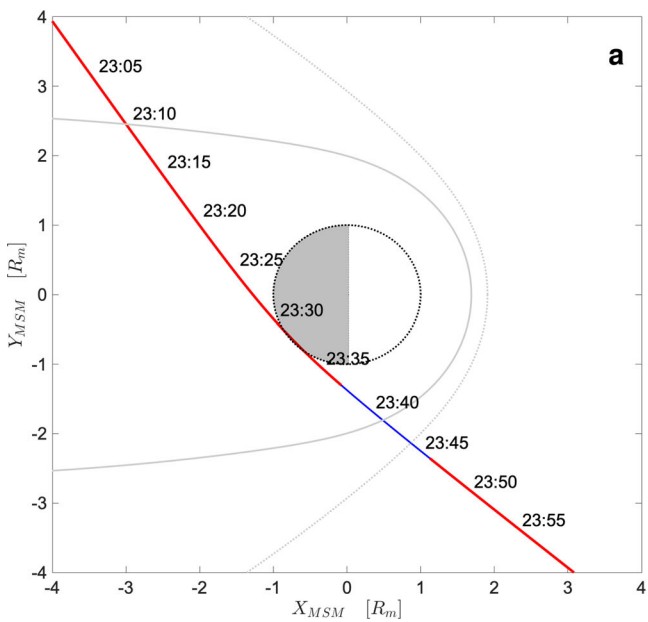
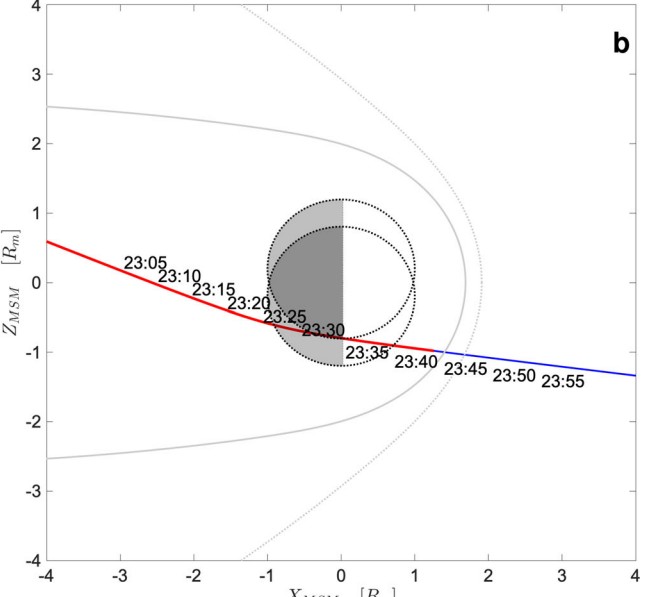

**Fig. 1 | Trajectory of BepiColombo during MFB1.** The trajectory of BepiColombo during the interval of interest, **a** in the $X_{MSM}$-$Y_{MSM}$ plane, **b** in the $X_{MSM}$-$Z_{MSM}$ plane. The solid gray line represents the magnetopause surface, while the dashed gray line correspond to the bow shock surface. Red lines correspond to the operational time of PICAM and MIPA sensors. Time labels are shown progressively along the trajectory. BC position data are given in the Source Datas file.

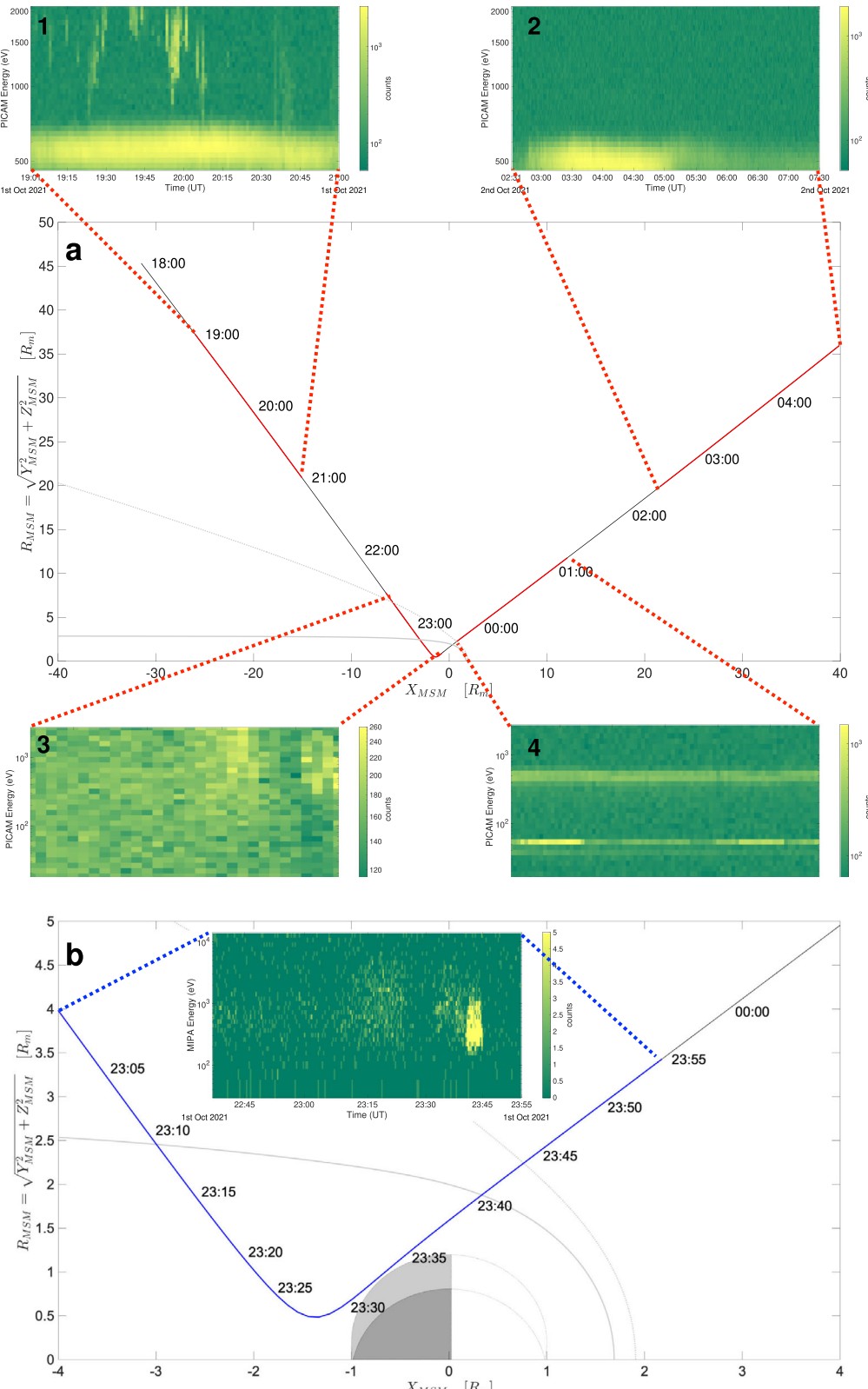

**Fig. 2 | Trajectory of BepiColombo and SERENA observations during MFB1.** The trajectory of BepiColombo during the interval of interest in the $X_{MSM}$-$R_{MSM}$ plane. **a** PICAM observations, **b** MIPA observations. The solid gray line represents the magnetopause surface, while the dashed gray line corresponds to the bow shock surface[25,26]. Red lines in **a** correspond to the operational time of PICAM, blue line in **b** corresponds to MIPA operation time. Time labels are shown progressively along the trajectory. Insets 1–4 in **a** show PICAM spectrograms for each specific time window, while the inset in **b** displays MIPA spectrograms. Color bars report ion counts in each specific time interval. BC position data are given in the Source Datas file.

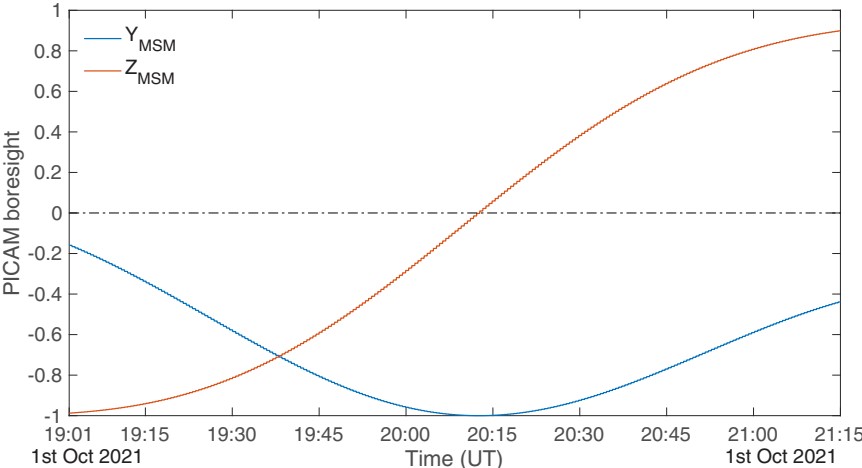

**Fig. 3 | PICAM boresight components along YMSM and ZMSM versus UT during MFB1.** The $Y_{MSM}$ and $Z_{MSM}$ components of the PICAM boresight are plootted versus UT for the time interval from 19:00 UT to 21:15 UT of the 1st of October, 2021. The blue line refers to the $Y_{MSM}$ component, while the red line refers to the $Z_{MSM}$ component. The horizontal dashed-dotted black line identifies the zero.

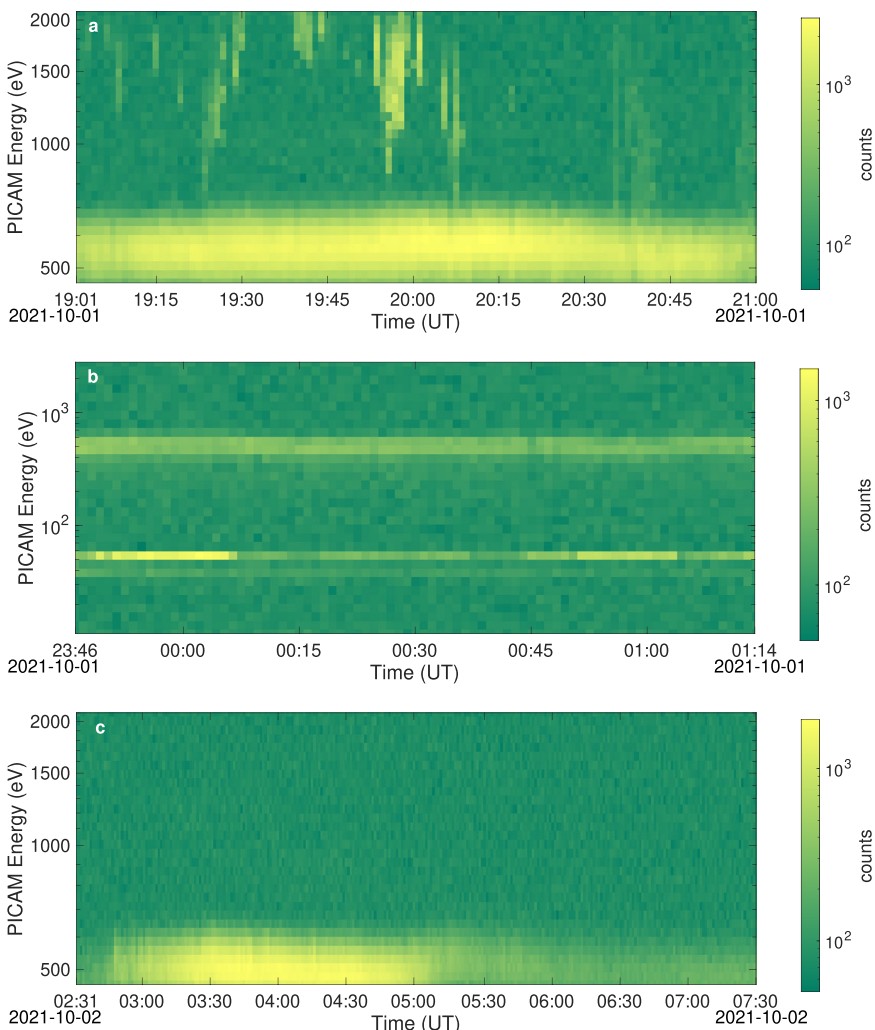

**Fig. 4 | MFB1 PICAM spectrograms outside Mercury's magnetosphere.** The spectrograms obtained by PICAM observations during the different time intervals: **a** from 19:01 UT to 21:00 UT on the 1st of October, 2021, **b** from 23:46 UT on the 1st of October, 2021, to 01:14 UT on the 2nd of October, 2021, **c** from 02:31 UT to 07:30 UT on the 2nd of October, 2021. Color bars report ion counts in each specific time interval.

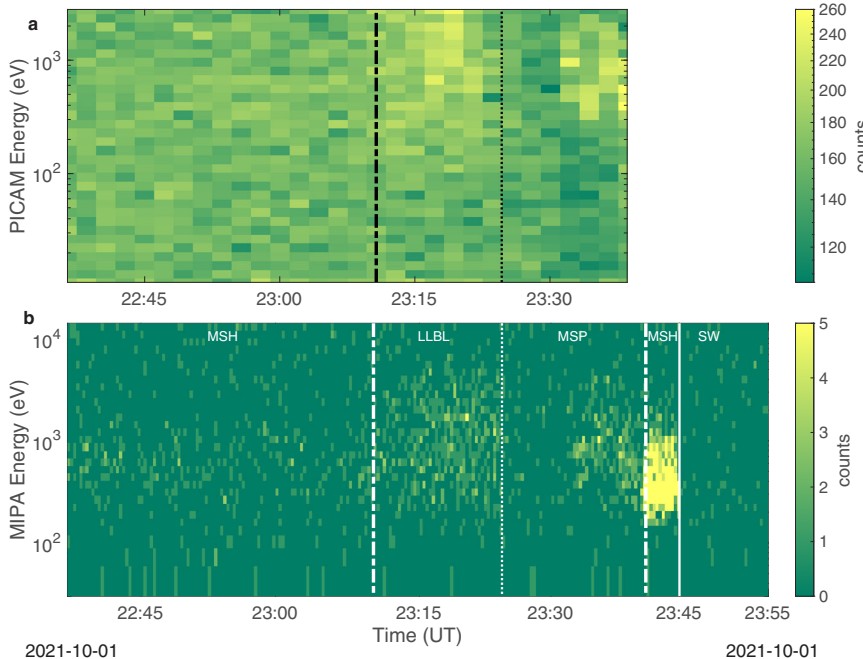

**Fig. 5 | MFB1 PICAM and MIPA spectrograms inside Mercury's magnetosphere.** The spectrograms obtained by **a** PICAM observations and **b** MIPA measurements in the inner magnetosphere of Mercury. The dashed-dotted lines refer to the expected inbound and outbound magnetopause crossings, the dotted lines refer to the observed transition from low latitude boundary layer to the magnetospheric dusk lobe, while the solid line in **b** marks the bow shock crossing. Labels refer to the different regions crossed by the spacecraft, specifically: magnetosheath (MSH), low latitude boundary layer (LLBL), inner magnetosphere (MSP), and solar wind (SW). Color bars report ion counts in each specific time interval.

However, as we will show in the next section the Mercury's magnetosphere was in quiet conditions, thus suggesting that it reconfigured after the passage of the flux rope. Such kind of events will be further investigated during the nominal mission (after satellite orbital insertions, in late 2025), when Mio will observe the solar wind conditions and simultaneously MPO will record any internal reaction. The solar wind observed upstream, on the dawn side of Mercury (Fig. 4c), shows a similar average energy, but appears to be more variable with a sharp drop in intensity after 5 UT, indicating an unstable condition. Just behind Mercury's bow shock, PICAM performed ion observations within an extended energy range, including lower energies. The solar wind energy was somewhat low, corresponding to about 550 eV (Fig. 4b). Two signals at even lower energies (the bands at 38 eV and 60 eV) were clearly observed, with a variable density on time scales of 30 min, with sunward and anti-sunward directions, respectively. Further investigation is needed by cumulating more events statistically significant with different environmental conditions and satellite orientations, in order to clarify whether this signal is originating from Mercury's interaction with the solar wind, or alternatively it is induced by spacecraft outgassing[11]. The Mass Spectrum Analyzer (MSA), a unit of the MPPE (Mercury Plasma Particle Experiment) consortium onboard BC-Mio, confirms the existence of a distinct double-band feature at low energies and that O + is the dominant ion species. The simultaneous observation by two separate BC instruments of such a low-energy signal excludes the possibility that it could come from instrumental effects. The persistent presence of outgassing material around spacecraft was discovered several years ago in the surrounding of Rosetta spacecraft[12]. In that case, a neutral gas cloud was actually discovered and the reason why such outgassing material was staying around the spacecraft is still not clearly understood. The possibility that the low-energy ion observations by BC could actually be determined by ionization and acceleration processes occurring on such a neutral gas cloud needs more investigations, so that several cruise campaigns have been planned to see when and in which conditions such a phenomenon is actually observed. Outbound from Mercury, the

about 550 eV slow solar wind is again observed when BC returned to cross the bow shock.

## 3. Magnetosheath and inner magnetosphere observations

The inbound bow shock crossing occurred before MIPA and PICAM were turned ON (after the wheel off-loading -WOL- operations). As shown in Fig. 5 (panel b), immediately after switch-on at 22:35 UT MIPA observed a weak signal at 800 eV–1 keV, corresponding to relatively hot magnetosheath population just barely observable within the MIPA FoV perpendicular to the Sun direction. As the spacecraft was moving upstream and closer to the planet, the ion temperature increased and a larger fraction of the distribution function was observed by both PICAM and MIPA (panels a and b). In fact, between about 23:10 UT and about 23:25 UT a signature of ion population was clearly observed by both PICAM and MIPA sensors as a wide distribution centered at about 1 keV. This population can be identified as low latitude boundary layer (LLBL)[13] similarly of what has been observed in the Earth magnetosphere[14], marking the transition between magnetosheath and magnetosphere. Just after this high density and hot signal at around 23:25 UT, the ion density decreased abruptly, possibly indicating that BC was inside the magnetosphere. At about 23:35 UT, the PICAM and MIPA ion intensity increased again (likely corresponding to the crossing of the plasma sheet), and simultaneously the PICAM background noise decreased significantly. This PICAM background noise decrease was observed also during the second Venus fly-by and it was interpreted as the shielding of galactic cosmic rays induced by the planet. Approaching the planet, where BC moved northward through the dawn flank plasma sheet, both PICAM and MIPA observed ions at energies between 300 eV and 2000 eV, just before the outbound magnetopause crossing occurred around 23:40 UT. Inside the magnetosphere, the only ion species clearly identified by PICAM is ionized hydrogen: further investigations are needed to identify possible presence of planetary ions in the data. This ion population could be the solar wind entered into the dayside magnetosphere and drifting clockwise around the planet viewed from the north rotational pole, i.e.,

ion grad B or curvature drift directions as with Earth's ring current (e.g. ref. [15]) and seen at higher altitudes by MESSENGER[18,19]. Approaching the dayside magnetopause at dawn, MIPA observed an increase in plasma ion densities and a decrease in the energy. This clear signature of dayside magnetosheath was registered only by MIPA between 23:40 UT and 23:45 UT (Fig. [5], panel b), while PICAM was switching its operation mode between 23:38 and 23:46 UT. In this observation, the magnetopause and bow shock crossings were registered at distances of 1.5 $R_M$ and 4 $R_M$, respectively, which is closer to the planet with respect to the average MESSENGER positions for these boundaries (Fig. [2]). The predicted crossing times for the bow shock and for the outbound magnetopause are about one to two minutes (note that the MIPA time resolution is 22 s) after the MFB1 observations by SERENA ion sensors.

## Discussion

In the present paper we report on the observation of the ion distributions in the environment of planet Mercury, at energies up to 15 keV, as detected by the sensors SERENA-PICAM and -MIPA, during the BC MFB1, on 1st October 2021.The data presented are ion observations in the southern hemisphere of the planet, down to an altitude of about 200 km, the closest approach during MFB1.The solar wind observed by SERENA before and after the magnetospheric crossing reveals the presence of a quite low-energy solar wind of about 500–600 eV. Moreover, we report the observation of intermittent events of high-energy solar wind pulses at about 1500 eV, which were observed during the inbound phase, far outside the bow shock, possibly due to the passage of an interplanetary flux rope. In addition, the outbound observation of the solar wind after the bow shock crossing revealed the presence of two beam-like signals at about 60 eV. This low-energy ion signal (which could be associated with satellite outgassing) is present in PICAM observations only outside the Mercury's Magnetosphere, and well separated from the higher energy solar wind signal. Hence, there is no indication that the observation of planetary plasma by PICAM could be affected by this phenomenon. Both the energetic spikes and the low-energy signals will be investigated in dedicated studies. Inside of Mercury's nightside magnetosphere, protons with energies of one to several keV are observed at low altitudes in the region where a weak ring current composed of drifting ions and electrons has been hypothesized[15,16]. These initial BC PICAM and MIPA data provide evidence for ring current-like distribution plasma around Mercury, as tentatively reported by MESSENGER data ([17], and reviews[18,19]). Further, the MIPA observations revealed a strong increase in plasma ion densities near the dawn magnetopause, slightly upstream of the terminator plane. Such increases in plasma beta (ratio of plasma thermal energy to magnetic energy) on the dawn side of Mercury's magnetosphere were also observed by MESSENGER during their flybys[20,21]. These new PICAM and MIPA observations appear to confirm the presence of this unexpected dayside magnetospheric asymmetry, tentatively reported by MESSENGER. Further analysis of the PICAM and MIPA measurements may lead the identification of its formation mechanism that is still eluded in the analyses of magnetosphere observations, e.g. double magnetopause[22], sunward transport of plasma sheet plasma[23] or a solar wind-driven low latitude boundary layer[13]. To summarize, SERENA ion sensors PICAM and MIPA detected various plasma regimes inside Mercury's magnetosphere, possibly allowing the identification of specific ion species and plasma populations, typical of plasma sheet, magnetosheath and magnetopause, up to the bow-shock crossing during the outbound phase. The relevance of these measurements emphasize the importance of the SERENA positive ion sensors. Once their data will be analyzed together with the MAG instrument magnetic field data and other instruments on board Mio and MPO, they will reveal important insight into many unknown aspects of a magnetosphere deep inside the inner heliosphere, like the case of Mercury. The observed plasma regions and features will be investigated in more detail by using new observations from the forthcoming five new Mercury flyby's and the nominal phases in Mercury's orbit starting in 2026[24].

## Data availability

The data referring to BC trajectory in Fig. 1, Fig. 2, and Fig. 3 are provided in the Source Data file. The SERENA raw data shown in Fig. 2, Fig. 4 and Fig. 5 are still in the proprietary period, due to BepiColombo data privacy regulations and cannot be distributed. Presently, these data may be only accessed via authorization in the SERENA team archive upon reasonable request to the SERENA team (PI, Stefano Orsini, stefano.orsini@inaf.it; or PI Deputy, Anna Milillo, anna.milillo@inaf.it). The data are expected to be available in the ESA PSA archive (https://archives.esac.esa.int/psa/#!Home%20View) before end of 2024. Source data are provided with this paper.

## Code availability

The codes related to the BC trajectory and to the SERENA raw data shown in Fig. 1, Fig. 2, Fig. 3, Fig. 4 and Fig. 5 are still in the proprietary period, due to BepiColombo data privacy regulations and cannot be distributed. Presently, these codes may be only accessed via authorization in the SERENA team archive upon reasonable request to the SERENA team (PI, Stefano Orsini, stefano.orsini@inaf.it; or PI Deputy, Anna Milillo, anna.milillo@inaf.it). The codes are expected to be available in the ESA PSA archive (https://archives.esac.esa.int/psa/#!Home%20View) before end of 2024.

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

## Acknowledgements

SERENA general management, System Control Unit (SCU) and Emitted Low Energy Neutral Atoms (ELENA) unit are funded by the Italian Space Agency (ASI) and by the Italian National Institute of Astrophysics (INAF), agreement n. 2018-8-HH.0. SERENA ground-based activity is also funded by the Expert support to SERENA Science Operations (EXPRO), ESA Contract Nr. C4000119196/16/ES/JD. Strofio unit is funded by NASA, through Marshall Space Flight Center under the Discovery Program Office. PICAM is funded mostly by the Austrian Space Appli9cations Programme (ASAP) of the Austrian Research Promotion Agency (FFG), and partially by the Programme de Dévelopement d'Expériences (PRO-DEX), and by the French Space Agency (CNES). MIPA is funded by the Swedish National Space Agency. Strofio and MIPA, as well as the general SERENA suite ground testing activities, have been also supported by the University of Bern, Switzerland.

## Author contributions

S. Orsini (S.O.), A. Milillo (A.Mi.), H. Lichtenegger (H.Li.), A. Varsani (A.Va.), S. Barabash (S.B.), S. Livi (S.L.), E. De Angelis (E.D.A.), G. Laky (G.L.), H. Nilsson (H.N.), M. Phillips (M.P.), A. Aronica (A.A.), E. Kallio (E.K.), P. Wurz (P.W.), T. Alberti (T.A.), A. Olivieri (A.O.), C. Plainaki (C.P.), J. A. Slavin (J.A.S.), I. Dandouras (I.D.), J. M. Raines (J.M.R.), J. Benkhoff (J.Be.), J. Zender (J.Z.), J.-J. Berthelier (J.-J.B.), M. Dosa (M.Do.), G. C. Ho (G.C.H.), R. M. Killen (R.M.K.), S. McKenna-Lawlor (S.M.-L.), K. Torkar (K.T.), O. Vaisberg (O.V.), F. Allegrini (F.A.), I. A. Daglis (I.A.D.), C. Dong (C.D.), C. P. Escoubet (C.P.E.), S. Fatemi (S.F.), M. Fränz (M.F.), S. Ivanovski (S.I.), H. Lammer (H.La.), François Leblanc (Fra.L.), V. Mangano (V.M.), A. Mura (A.Mu.), R. Risfpoli (R.R.), M. Sarantos (M.S.), H. T. Smith (H.T.S.), M. Wieser (M.W.), F. Camozzi (F.C.), A. M. Di Lellis (A.M.D.L.), G. Fremuth (G.F.), F. Giner (F.G.), R. Gurnee (R.G.), J. Hayes (J.H.), H. Jeszenszky (H.J.), B. Trantham (B.T.), J. Balaz (J.Ba.), W. Baumjohann (W.B.), M. Cantatore (M.C.), D. Delcourt (D.D.), M. Delva (M.Del.), M. Desai (M.Des.), H. Fischer (H.F.), A. Galli (A.G.), M. Grande (M.G.), M. Holmström (M.H.), I. Horvath (I.H.), K.C. Hsieh (K.C.H.), R. Jarvinen (R.J.), R. E. Johnson (R.E.J.), A. Kazakov (A.K.), K. Kecskemety (K.K.), H. Krüger (H.K.), C. Kürbisch (C.K.), Frederic Leblanc (Fre.L.), M. Leichtfried (M.L.), E. Mangraviti (E.M.), S. Massetti (S.M.), D. Moissenko (D.M.), M. Moroni (M.M.), R. Noschese (R.N.), F. Nuccilli (F.N.), N. Paschalidis (N.P.), J. Ryno (J.R.), K. Seki (K.Se.), A. Shestakov (A.S.), S. Shuvalov (S.Sh.), R. Sordini (R.S.), F. Stenbeck (F.S.), J. Svensson (J.S.), S. Szalai (S.Sz.), K. Szego (K.Sz.), D. Toublanc (D.T.), N. Vertolli (N.V.), R. Wallner (R.W.), A. Vorburger (A.Vo.). Conceptualization: S.O., A.Mi., H.Li., A.Va., S.B., S.L.; methodology: S.O., A.Mi., H.Li., A.Va., S.B., S.L., A.A.; investigation: S.O., A.Mi., A.Vo., S.B., E.D.A., G.L., H.N., E.K., P.W., T.A., J.A.S., J.M.R., I.D.; visualization: S.O., A.Mi., A.Va., S.B., E.D.A., E.K., J.A.S., A.A.; funding acquisition: A.O., S.O., A.Mi., H.Li., A.Va., S.B., S.L.; project administration: A.O., S.O., A.Mi., H.Li., A.Va., S.B., S.L.; supervision: S.O.; writing original draft: S.O., A.Mi., H.Li., A.Va., S.B., S.L., E.D.A., G.L., H.N., M.P., A.A., E.K., P.W., T.A., A.O., C.P., J.A.S., I.D., J.M.R.; writing review and editing: S.O., A.Mi., H.Li., A.Va., S.B., S.L., E.D.A., G.L., H.N., M.P., A.A. E.K., P.W., T.A., A.O., C.P., J.A.S., I.D., J.M.R., J.Be., J.Z., J.-J.B., M.Do., J.H., R.M.K., S.M.-L., K.T., O.V., F.A., I.A.D., C.D., C.P.E., S.F., M.F., S.I., H.La., Fra.L., V.M., A.Mu., R.R., M.S., H.T.S., M.W., F.C., A.M.D.L., G.F., F.G., R.G., J.H., H.J., B.T., J.Ba., W.B., M.C., D.D., M.Del., M.Des., H.F., A.G., M.G., M.H., I.H., K.C.H., R.J., R.E.J., A.K., K.K., H.K., C.K., Fre.L., M.L., E.M., S.M., D.M., M.M., R.N., F.N., N.P., J.R., K.Se. A.S., S.Sh., R.S., F.S., J.S., S.Sz., K.Sz., D.T., N.V., R.W., A.Vo.

## Competing interests

The authors declare no competing interests.

## Additional information

S. Orsini ®[1] ✉, A. Milillo ®[1], H. Lichtenegger[2], A. Varsani ®[2], S. Barabash[3], S. Livi[4,5], E. De Angelis[1], T. Alberti[1], G. Laky[2], H. Nilsson[3], M. Phillips ®[4], A. Aronica ®[1], E. Kallio ®[6], P. Wurz ®[7], A. Olivieri ®[8], C. Plainaki[8], J. A. Slavin ®[5], I. Dandouras ®[9], J. M. Raines[5], J. Benkhoff ®[10], J. Zender[10], J.-J. Berthelier[11], M. Dosa[12], G. C. Ho ®[13], R. M. Killen ®[14], S. McKenna-Lawlor[15], K. Torkar[2], O. Vaisberg[16], F. Allegrini[4,17], I. A. Daglis ®[18,19], C. Dong ®[20], C. P. Escoubet[10], S. Fatemi ®[21], M. Fränz[22], S. Ivanovski[23], N. Krupp[22], H. Lammer[2], François Leblanc[11], V. Mangano ®[1], A. Mura[1], R. Rispoli ®[1], M. Sarantos[14], H. T. Smith[13], M. Wieser ®[3], F. Camozzi[24], A. M. Di Lellis[25], G. Fremuth[2], F. Giner[2], R. Gurnee[26], J. Hayes[13], H. Jeszenszky[2], B. Trantham[2], J. Balaz ®[27], W. Baumjohann[2], M. Cantatore ®[24], D. Delcourt[28], M. Delva[2], M. Desai[4], H. Fischer[22], A. Galli ®[7], M. Grande ®[29], M. Holmström[3], I. Horvath[12], K. C. Hsieh[30], R. Jarvinen ®[6,31], R. E. Johnson[32], A. Kazakov ®[1], K. Kecskemety[12], H. Krüger[22], C. Kürbisch[2], Frederic Leblanc[33], M. Leichtfried[2], E. Mangraviti[23], S. Massetti ®[1], D. Moissenko ®[16], M. Moroni[1], R. Noschese[1], F. Nuccilli[1], N. Paschalidis[14], J. Ryno ®[31], K. Seki ®[34], A. Shestakov[16], S. Shuvalov ®[16], R. Sordini ®[1], F. Stenbeck[3], J. Svensson[3], S. Szalai[12], K. Szego[12,35], D. Toublanc[9], N. Vertolli[1], R. Wallner[2] & A. Vorburger[7]

[1]Institute of Space Astrophysics and Planetology, INAF, Roma, Italy. [2]Space Research Institute, Austrian Academy of Sciences, Graz, Austria. [3]Swedish Institute of Space Physics, Kiruna, Sweden. [4]Southwest Research Institute, San Antonio, TX, USA. [5]University of Michigan, Department of Climate and Space Sciences and Engineering, Ann Arbor, MI, USA. [6]Aalto University, Department of Electronics and Nanoengineering, School of Electrical Engineering, Helsinki, Finland. [7]University of Bern, Institute of Physics, Bern, Switzerland. [8]Italian Space Agency, ASI, Roma, Italy. [9]Institut de Recherche en Astrophysique et Planétologie, CNRS, CNES, Université de Toulouse, Toulouse, France. [10]ESA-ESTEC, Noordwijk, The Netherlands. [11]LATMOS/IPSL, CNRS, Sorbonne Université, Paris, France. [12]Wigner Research Centre for Physics, Budapest, Hungary. [13]The Johns Hopkins University Applied Physics Laboratory, Laurel, MD 20723, USA. [14]NASA/Goddard Space Flight Center, Greenbelt, MD 20771, USA. [15]Space Technology Ireland, Ltd., Maynooth, Co., Kildare, Ireland. [16]IKI Space Research Institute, Moscow, Russia. [17]University of Texas at San Antonio, Department of Physics and Astronomy, San Antonio, TX, USA. [18]National and Kapodistrian University of Athens, Department of Physics, Athens, Greece. [19]Hellenic Space Center, Athens, Greece. [20]Princeton Plasma Physics Laboratory and Department of Astrophysical Sciences, Princeton University, Princeton, NJ, USA. [21]Department of Physics, Umeå University, Umeå, Sweden. [22]Max-Planck-Institut für Sonnensystemforschung, MPS, 37077 Göttingen, Germany. [23]Astronomincal Observatory, INAF, Trieste, Italy. [24]OHB-Italia SpA, Milano, Italy. [25]AMDL srl, Roma, Italy. [26]Laboratory for Atmospheric and Space Physics, Boulder, CO, USA. [27]Institute of Experimental Physics SAS, Slovak Academy of Sciences, 040 01 Košice, Slovakia. [28]Universitè d'Orleans, Orleans, France. [29]Aberystwyth University, Aberystwyth, Ceredigion, UK. [30]University of Arizona, Tucson, AZ, USA. [31]Finnish Meteorological Institute FMI, Helsinki, Finland. [32]University of Virginia, Charlottesville, VA 22904, USA. [33]LPP, École polytechnique, 91128 Palaiseau Cedex, Paris, France. [34]University of Tokyo, Department of Earth and Planetary Science, Graduate School of Science, Tokyo, Japan. [35]Deceased: Karoly Szego ✉e-mail: stefano.orsini@inaf.it

