## [Peer Review File · Nature Communications]

Review of NCOMM Manuscript 22-11596-T:

First Observations of Mercury's Inner Southern Magnetosphere by BepiColombo/SERENA Ion Sensors

S. Orsini et al.

This paper presents initial results from the first flyby of Mercury from the BepiColombo spacecraft. In particular, ion energy spectra are shown from upstream of the bow shock, inbound crossing of the bow shock and magnetopause, inside the magnetosphere and then outbound across the magnetopause and bow shock again. Many of the results confirm and agree with those observed by MESSENGER in the magnetopause and magnetosphere, although there are some interesting new findings of higher energy ions upstream of the bow shock on the inbound and lower energy heavy ions on the outbound. Also, observations are presented from the inner southern magnetosphere, a region not observed by MESSENGER. Overall, the manuscript is well written, nicely organized, includes interesting new results and I recommend publication after some minor text alterations listed below.

There is one question concerning the statement on page 4 that “O⁺ is the dominant ion species”. Is this definitive or could it possibly be H₂O⁺ water group ions or Na⁺ sodium group ions that have commonly been observed by MESSENGER in the vicinity of Mercury? If possible, the authors might want to slightly expand this part of the discussion of heavy ions observed near Mercury.

List of suggested grammatical and typographical changes:

Page 2 **Main Text:**

Lines 3, 4: change “MIO still needs a long time (at the end of 2025)” to “MIO will take place at the end of 2025.”

Line 5: change “one just occurred” to “one that just occurred”

Line 16: replace “inside the Mercury magnetosphere.” with “inside Mercury’s magnetosphere.”

Line 18: change last sentence to read “The trajectory of the first Mercury flyby (MFB1) covers regions in the southern hemisphere at low altitudes not explored by previous missions, and the data presented show for the first time ion energy distributions at the bow shock and closer to Mercury in the southern hemisphere.”

Page 2 BepiColombo trajectory and region traversals.

Line 7: change “MESSENGER errors” to “MESSENGER error”

Line 8: change “orbital velocity completes” to “orbital velocity, which completes”

Line 13: switch “crossing again” to “again crossing”

Line 16: change “upstream the bow shock” to “upstream of the bow shock”

Line 17: change “observed magnetosheath” to “observed the magnetosheath”

Page 4 Solar wind observations

Line 15: change “that this signal was never observed” to “that such a signal was not observed”

Line 17: change “We wonder” to “It remains to be determined”

Line 20: change “variable in intensity, indicating an unstable condition.” to “variable with a sharp drop in intensity after ~ 5:00 UT.”

Line 25: change “will be” to “are”

Lines 31, 32: change “As a matter of fact, the Mass...” to “The Mass...”.

Page 6 Magnetosheath and inner magnetosphere observations.

Line 4: change “to relatively hot” to “to a relatively hot”

Line 7: change “a signature of ion population” to “an ion population signature”

Line 17: delete one of the commas after “plasma sheet,,”

Lines 24 and 29 the acronym “MSG” is used which has not been previously defined; presumably this stands for MESSENGER. Either MSG needs to be defined or perhaps its better just to include MESSENGER in full each time.

Line 26: change “of dayside magnetosheath” to “of the dayside magnetosheath”

Line 31 (last line): replace “well in” with “in good”

Page 8 Conclusions

Line 11: replace “dedicated” with “future”

Line 15: put “a” before “ring current”

Lines 16, 19, 22: use of “MSG” (see previous page 6 comments)

Lines 22, 23: Acronym “MSP” appears for first time, needs to be defined or spelled out in full.

Line 27: put “the” before “plasma sheet”

Line 30: replace “will be” with “is”.

Line 32: delete repeated phrase “many unknown aspects”

Line 34: replace “forthcoming” with “upcoming” and delete “new” before “Mercury”

Line 35 (last line): replace “flyby’s” with “flybys” (no apostrophe)

Page 9 **References:**

Reference 8. Slavin, et al.: replace “Magnetosphe” with “Magnetosphere”

Supplementary Material

Page 1 Data Downlink and processing data pipeline:

Line 4: change “where is ready” to “where it is ready”

Line 6: change “is a binary data” to “is binary data”

Line 8: near end: should be spelled “Spacecraft”

Line 9: first word should be spelled “Elapsed”

Line 28: “All” should be “all”

Page 2 MPIA (Miniature Ion Precipitation Analyzer) ← **add closing parenthesis to title**

Line 7: change “does not have own data” to “does not have its own data”

Line 11 (last line): change “used sufficient” to “used, which is sufficient” and change “heavies” to “heavy ions”

Figure S2 caption: put “is” before “decoupled” and “a” before “titanium”

Table S1: in the right half of the table for **Science region**, change the table borders such that “Magnetosphere” appears on one line.

Reviewer #2 (Remarks to the Author):

Review of 'First observations of Mercury's inner southern magnetosphere by BepiColombo/SERENA ion sensors'

By S. Orsini, et al.

This paper presents the first observations of Mercury's inner southern magnetosphere and nearby regions, as measured by Bepi-Colombo sensors SERENA-PICAM and -MIPA, during the first Mercury flyby on October 1, 2021. More specifically, the authors present ion observations in the southern hemisphere of the planet's magnetosphere, down to an altitude of ~ 200 km. In particular, this flyby took place under relatively low-energy solar wind conditions (~ 500 - 600 eV). The authors also report observations of an intermittent event at high energy in the solar wind (~ 1.5 keV), far outside the shock in the dusk region. Finally, observations gathered during the outbound leg reveal the presence of two beam-like signals at about 60 eV.

The first measurements provided by SERENA ion sensors PICAM and MIPA are certainly interesting and relevant and show the importance these instruments have for the Bepi-Colombo mission and future investigation of the Hermean magnetosphere. My main concern with this paper is that neither the nature of the energetic spike nor the low-energy signals are investigated in detail and are left for future studies. Moreover, this paper would certainly benefit from magnetic field observations, especially if the energetic spike may be a foreshock feature. However, this also seems to be left for a future investigation. In addition, the authors report that the two beam-like signals at about 60 eV may be induced by spacecraft outgassing, thus raising the question: could this signature be instrumental? I think this is important since this does not seem to be a weak signal. In particular, the number of counts is much higher than that of the solar wind, at least between $\sim 23:49$ and $00:05$ UT (Figure 2B).

A similar concern is present for the high-energy population displayed in Figure 2A. It is reported that this signature could be associated with foreshock ions. Do the authors have any estimation up to which distance foreshock ions could travel given the conditions around Mercury and the relatively small bow shock compared to the terrestrial counterpart? Is there any explanation for the observed intermittency? Could this be due to changes in the spacecraft/instrument orientation? Is the energy ratio between the high-energy and the solar wind ions in agreement with expectations from particle acceleration at the bow shock? Is the particle flux ratio between high-energy ions and the solar wind in agreement with expectations from particle acceleration at the bow shock?

In addition, regarding Figure 3B, what is the minimum number of counts above which MIPA observations are significant? Most of the energy bins present less than two counts.

Because of these reasons, I am afraid I cannot recommend publication in Nature Communications in the present form. Please find below my comments for your consideration.

Specific comments:

Please define all acronyms used in this manuscript. For instance, Page 2, in the Main Text section: MPO and MIO are not defined. Also, please avoid using acronyms when possible. For instance, MSP is used only twice and it is not defined.

Page 2: 'The dipole is offset northward along the MSM Z-axis by 480 km (~ 0.2 RM) and parallel to the planetary rotation axis and anti-parallel to the magnetic dipole to within the MESSENGER errors of ~ 1 deg (4).' This sentence is not clear. Please rewrite.

Page 2: 'The spacecraft approached the planet from the dusk flank, the magnetosheath and near magnetotail, and exited the magnetosphere in the dawn dayside, crossing again the magnetosheath.' It could be worth adding two panels to Figure 1 displaying the trajectory of Bepi-Colombo in the X-Y and X-Z MSM planes.

Page 2: 'These coordinates are often aberrated using the measured or estimated solar wind speed such that the radial solar wind velocity points opposite to the MSM X' direction, where the prime symbol indicates that the X and Y axes have been aberrated (e.g.: 5).' The manuscript refers to aberrated coordinates that are not used in the manuscript. Please justify why this coordinate system was not considered but was mentioned in this line.

Figures 1 and 2: I think the resolution of both figures should be significantly improved.

Page 2: 'As shown in Figure 1, PICAM operated during 4 distinct time periods and observed the downstream and upstream solar wind ion flux (panel A, inserts 1 and 4)'. Inserts 1 and 4 seem to show plasma data upstream from the bow shock. What does the term 'downstream' mean here?

Page 4: 'Between 19:00 UT and 21:00 UT, at a distance of about 25 RM from Mercury center, in the dusk side, the spacecraft rotated and the PICAM boresight moved from the -ZMSM direction, i.e., the southern hemisphere to +ZMSM in the northern hemisphere. In doing so, PICAM FoV passed through the -YMSM direction (i.e., moving to the same direction as the planet moves pointing toward the bow shock)'. Was this rotation present over all this time interval? It would be worth adding a figure showing how that spacecraft and the field of view of the instrument changed with time. Thus, the reader can more easily associate changes in the observed signatures with variability in the field of view.

Page 4: 'The possibility that these dynamic events are foreshock ions (e.g.: 7), transported along the magnetic field spiral, will be investigated in a future study' Can the authors justify this interpretation? Can foreshock ions reach these distances, despite the relatively small bow shock of Mercury? Was the Interplanetary Magnetic Field orientation during this event consistent with this interpretation? Reference (7) performed a global hybrid simulation of the ion foreshock relatively close to the bow shock, compared with the distances where the intermittent signature present in Figure 2A is observed. Can the authors rule out an instrumental component in this observation?

Page 4: 'A more detailed analysis of these features will be performed when other BC data, such as magnetic field measurements will be available' In my opinion, magnetic field data is necessary to provide information regarding Bepi-Colombo connectivity to the bow shock when detecting the high energy intermittent signal.

Page 4: 'However, we must note that this signal was never observed in the extensive plasma ion measurements recorded by the MESSENGER mission, which orbited Mercury from 2011 to 2015.' What do the authors mean by 'this signal' in this sentence? What is the origin of this signal? It is my understanding that foreshock ions have been observed by MESSENGER FIPS. This is reported, for instance, in Tracy, Patrick J. PhD Thesis (2016) <https://ui.adsabs.harvard.edu/abs/2016PhDT.....116T> and was also presented at AGU fall meetings <https://ui.adsabs.harvard.edu/abs/2018AGUFMSM51B..02G>

Page 4: 'Just behind Mercury's bow shock, PICAM performed ion observations within an extended energy range, including lower energies. The solar wind speed was somewhat low at about 300 km/s corresponding to ~ 550 eV (Figure 2B). Two signals at even lower energies (the bands at 38 eV and 60 eV) were clearly observed, with a variable density on time scales of 30 minutes, with sunward and anti-sunward directions, respectively. Further investigation is needed to clarify whether this signal is originating from Mercury's interaction with the solar wind or if they are induced by spacecraft outgassing (9, 10).' Could this signature be instrumental? I think this is important since this does not seem to be a weak signal. In particular, the number of counts is much higher than that of the solar wind, at least between ~23:49 and 00:05 UT (Figure 2B).

Page 6 'Just after this high density and hot signal, an abrupt change of plasma conditions seem to indicate the magnetopause crossing.' What is the distance between the observed and the expected magnetopause location based on (6)?

Page 6 'This PICAM background noise decrease was observed also during the second Venus flyby and it was interpreted as the shielding of galactic cosmic rays induced by the planet.' Please add a reference.

Page 6 '...northward through the dawn flank plasma sheet,, both PICAM and MIPA...' Please remove a comma.

Page 6 '...just before the outbound magnetopause crossing occurred around 23:40 UT.' I would add a vertical dotted line showing the location of the observed outbound magnetopause crossing.

Page 8: '... they will reveal important insight into many unknown aspects many unknown aspects of a magnetosphere deep inside the inner heliosphere, like the case of Mercury...' Many unknown aspects is repeated, please correct.

Reviewer #3 (Remarks to the Author):

This manuscript provided first observations of Mercury's *inner magnetosphere made by the SERENA instrument package onboard the Bepi-Colombo spacecraft. This is also the first observations of Mercury's southern part of the magnetosphere, which was previously un-observed by the MESSENGER mission and not studied extensively by earlier studies. Hence this manuscript provided a first look at the possibilities results from the Bepi-Colombo mission can offer, and this work is of significance to future Mercury science.

This manuscript is good as it is. The methodology, analysis and interpretation of the ion measurements of the first Mercury flyby by PICAM and MIPA is straightforward and sound. I have no major comments regarding the results presented in this manuscript.

Minor comment:

To help the reader better understand Figure 3, I would highly recommend the authors labelled in the figure itself the different regions of Mercury's magnetosphere that each lines in Figure 3b represents. It can be confusing to have to constantly refer to the figure caption to understand which magnetospheric region did Bepi-Colombo observed. Alternatively, the use of different colors should also be considered.

Review of 'First observations of Mercury's inner southern magnetosphere by BepiColombo/SERENA ion sensors', by S. Orsini, et al.

Reply to Reviewer #1

General comment

- *"I support publication of this manuscript. See attached document with the detailed review."*

"This paper presents initial results from the first flyby of Mercury from the BepiColombo spacecraft. In particular, ion energy spectra are shown from upstream of the bow shock, inbound crossing of the bow shock and magnetopause, inside the magnetosphere and then outbound across the magnetopause and bow shock again. Many of the results confirm and agree with those observed by MESSENGER in the magnetopause and magnetosphere, although there are some interesting new findings of higher energy ions upstream of the bow shock on the inbound and lower energy heavy ions on the outbound. Also, observations are presented from the inner southern magnetosphere, a region not observed by MESSENGER. Overall, the manuscript is well written, nicely organized, includes interesting new results and I recommend publication after some minor text alterations listed below.

List of suggested grammatical and typographical changes:

Page 2 Main Text:

Lines 3, 4: change "MIO still needs a long time (at the end of 2025)" to "MIO will take place at the end of 2025."

Line 5: change "one just occurred" to "one that just occurred"

Line 16: replace "inside the Mercury magnetosphere." with "inside Mercury's magnetosphere."

Line 18: change last sentence to read "The trajectory of the first Mercury flyby (MFB1) covers regions in the southern hemisphere at low altitudes not explored by previous missions, and the data presented show for the first time ion energy distributions at the bow shock and closer to Mercury in the southern hemisphere."

Page 2 BepiColombo trajectory and region traversals.

Line 7: change "MESSENGER errors" to "MESSENGER error"

Line 8: change "orbital velocity completes" to "orbital velocity, which completes"

Line 13: switch "crossing again" to "again crossing"

Line 16: change "upstream the bow shock" to "upstream of the bow shock"

Line 17: change "observed magnetosheath" to "observed the magnetosheath"

Page 4 Solar wind observations

Line 15: change "that this signal was never observed" to "that such a signal was not observed"

Line 17: change "We wonder" to "It remains to be determined"

Line 20: change "variable in intensity, indicating an unstable condition." to "variable with a sharp drop in intensity after ~ 5:00 UT."

Line 25: change "will be" to "are"

Lines 31, 32: change "As a matter of fact, the Mass..." to "The Mass..."

Page 6 Magnetosheath and inner magnetosphere observations.

Line 4: change "to relatively hot" to "to a relatively hot"

Line 7: change "a signature of ion population" to "an ion population signature"

Line 17: delete one of the commas after "plasma sheet,"

Lines 24 and 29 the acronym "MSG" is used which has not been previously defined; presumably this

stands for MESSENGER. Either MSG needs to be defined or perhaps it's better just to include MESSENGER in full each time.

Line 26: change "of dayside magnetosheath" to "of the dayside magnetosheath"

Line 31 (last line): replace "well in" with "in good"

Page 8 Conclusions

Line 11: replace “dedicated” with “future”
Line 15: put “a” before “ring current”
Lines 16, 19, 22: use of “MSG” (see previous page 6 comments)
Lines 22, 23: Acronym “MSP” appears for first time, needs to be defined or spelled out in full.
Line 27: put “the” before “plasma sheet”
Line 30: replace “will be” with “is”.
Line 32: delete repeated phrase “many unknown aspects”
Line 34: replace “forthcoming” with “upcoming” and delete “new” before “Mercury”
Line 35 (last line): replace “flyby’s” with “flybys” (no apostrophe)
Page 9 References:
Reference 8. Slavin, et al.: replace “Magnetosphe” with “Magnetosphere”
Supplementary Material
Page 1 Data Downlink and processing data pipeline:
Line 4: change “where is ready” to “where it is ready”
Line 6: change “is a binary data” to “is binary data”
Line 8: near end: should be spelled “Spacecraft”
Line 9: first word should be spelled “Elapsed”
Line 28: “All” should be “all”
Page 2 MPIA (Miniature Ion Precipitation Analyzer) ← add closing parenthesis to title
Line 7: change “does not have own data” to “does not have its own data”
Line 11 (last line): change “used sufficient” to “used, which is sufficient” and change “heavies” to “heavy ions”
Figure S2 caption: put “is” before “decoupled” and “a” before “titanium”
Table S1: in the right half of the table for Science region, change the table borders such that “Magnetosphere” appears on one line.”

We gratefully thank Reviewer #1 for approving publication of this paper with minor comments, which have been fully addressed in the revision (see revisions listed in the Office word file).

There is one question concerning the statement on page 4 that “O⁺ is the dominant ion species”. Is this definitive or could it possibly be H₂O⁺ water group ions or Na⁺ sodium group ions that have commonly been observed by MESSENGER in the vicinity of Mercury? If possible, the authors might want to slightly expand this part of the discussion of heavy ions observed near Mercury.

At present, we think that it is a spacecraft-induced effect, and we have not yet considered the possible composition of outgassing material from the spacecraft. Nevertheless, in the revised version we now mention similar outgassing features in other observations around Rosetta spacecraft, and claim for further studies and statistics for better understand this important matter. In the following we show our revised text (line 157-169):

“The Mass Spectrum Analyzer (MSA), a unit of the MPPE consortium onboard BC-MIO, confirms the existence of a distinct double-band feature at low energies and that O⁺ is the dominant ion species (Lina Hadid, private communication). The simultaneous observation by two separate BC instruments of such a low energy signal excludes the possibility that it could come from instrumental effects. The persistent presence of outgassing material around spacecraft was discovered several years ago in the surrounding of Rosetta spacecraft (11). In that case a neutral gas cloud was actually discovered and the reason why such outgassing material was staying around the spacecraft without evaporating in space suggested the idea of a sort of potential barrier causing recurrent ionization and neutralization processes. The very fact that in the case of BC such an ion potential is actually determining the existence of two distinct signals will certainly need more investigations, so that many other cruise

campaigns have been planned to see when and in which conditions such a phenomenon is actually observed”.

Reply to Reviewer #2

General comment

- *“This paper presents the first observations of Mercury’s inner southern magnetosphere and nearby regions, as measured by Bepi-Colombo sensors SERENA-PICAM and –MIPA, during the first Mercury flyby on October 1, 2021. More specifically, the authors present ion observations in the southern hemisphere of the planet’s magnetosphere, down to an altitude of ~200 km. In particular, this flyby took place under relatively low-energy solar wind conditions (~500-600 eV). The authors also report observations of an intermittent event at high energy in the solar wind (~1.5 keV), far outside the shock in the dusk region. Finally, observations gathered during the outbound leg reveal the presence of two beam-like signals at about 60 eV. The first measurements provided by SERENA ion sensors PICAM and MIPA are certainly interesting and relevant and show the importance these instruments have for the Bepi-Colombo mission and future investigation of the Hermean magnetosphere. My main concern with this paper is that neither the nature of the energetic spike nor the low-energy signals are investigated in detail and are left for future studies. Moreover, this paper would certainly benefit from magnetic field observations, especially if the energetic spike may be a foreshock feature. However, this also seems to be left for a future investigation.*

We very much appreciate the effort of the Reviewer#2 to help us make this paper more focused to its goals. We have done our best to address all of his/her comments. Here below we face the solicited point, doing our best for addressing each of them (see the review as shown in the revision word file.

In addition, the authors report that the two beam-like signals at about 60 eV may be induced by spacecraft outgassing, thus raising the question: could this signature be instrumental? I think this is important since this does not seem to be a weak signal. In particular, the number of counts is much higher than that of the solar wind, at least between ~23:49 and 00:05 UT (Figure 2B).

We have extensively commented the low-energy ion observation within the point-to-point response section.

A similar concern is present for the high-energy population displayed in Figure 2A. It is reported that this signature could be associated with foreshock ions. Do the authors have any estimation up to which distance foreshock ions could travel given the conditions around Mercury and the relatively small bow shock compared to the terrestrial counterpart? Is there any explanation for the observed intermittency? Could this be due to changes in the spacecraft/instrument orientation? Is the energy ratio between the high-energy and the solar wind ions in agreement with expectations from particle acceleration at the bow shock? Is the particle flux ratio between high-energy ions and the solar wind in agreement with expectations from particle acceleration at the bow shock?

The interpretation of this high energy signal has been revised. Details are given in the point-to-point response section.

In addition, regarding Figure 3B, what is the minimum number of counts above which MIPA observations are significant? Most of the energy bins present less than two counts.

The reviewer is correct: MIPA counts in the range of 1-2 counts should not be taken into consideration. Nevertheless, we take profit from the more sensitive PICAM data

to identify some of the magnetospheric regions, as encountered during this fly-by. Actually, MIPA data are very useful in the last part of the plots, where PICAM observations stopped, whereas MIPA was still operating. We have added a short note in the caption of Figure 4. (lines 267-271).

Because of these reasons, I am afraid I cannot recommend publication in Nature Communications in the present form. Please find below my comments for your consideration."

Point-to-point responses

- *"Please define all acronyms used in this manuscript. For instance, Page 2, in the Main Text section: MPO and MIO are not defined. Also, please avoid using acronyms when possible. For instance, MSP is used only twice and it is not defined."*

MSP stays for Magnetosphere. We have excluded or explained the acronyms as requested throughout the whole manuscript.

- *"Page 2: 'The dipole is offset northward along the MSM Z-axis by 480 km (~ 0.2 RM) and parallel to the planetary rotation axis and anti-parallel to the magnetic dipole to within the MESSENGER errors of ~ 1 deg (4).' This sentence is not clear. Please rewrite."*

Changed (line 101-102):

"The dipole is offset northward along the MSM Z-axis by 480 km (~ 0.2 RM), parallel to the planetary rotation axis (4).

- *"Page 2: 'The spacecraft approached the planet from the dusk flank, the magnetosheath and near magnetotail, and exited the magnetosphere in the dawn dayside, crossing again the magnetosheath.' It could be worth adding two panels to Figure 1 displaying the trajectory of Bepi-Colombo in the X-Y and X-Z MSM planes."*

We have added two panels to Figure 1 as suggested by the reviewer.

- *"Page 2: 'These coordinates are often aberrated using the measured or estimated solar wind speed such that the radial solar wind velocity points opposite to the MSM X' direction, where the prime symbol indicates that the X and Y axes have been aberrated (e.g.: 5).' The manuscript refers to aberrated coordinates that are not used in the manuscript. Please justify why this coordinate system was not considered but was mentioned in this line."*

The reviewer is right and to avoid confusion we deleted this reference to the aberrate system. (line 103)

- *"Figures 1 and 2: I think the resolution of both figures should be significantly improved."*

We improved the quality accordingly (using a 300 dpi resolution). However, the main problem is the encapsulated procedure via the MSword. This issue will be fixed by the Production Office.

- *"Page 2: 'As shown in Figure 1, PICAM operated during 4 distinct time periods and observed the downstream and upstream solar wind ion flux (panel A, inserts 1 and 4)'. Inserts 1 and 4 seem to show plasma data upstream from the bow shock. What does the term 'downstream' mean here?"*

The reviewer is right. We deleted the term "downstream", as well as "upstream" (line 104-105)

- *“Page 4: ‘Between 19:00 UT and 21:00 UT, at a distance of about 25 RM from Mercury center, in the dusk side, the spacecraft rotated and the PICAM boresight moved from the –ZMSM direction, i.e., the southern hemisphere to +ZMSM in the northern hemisphere. In doing so, PICAM FoV passed through the –YMSM direction (i.e., moving to the same direction as the planet moves pointing toward the bow shock)’. Was this rotation present over all this time interval? It would be worth adding a figure showing how that spacecraft and the field of view of the instrument changed with time. Thus, the reader can more easily associate changes in the observed signatures with variability in the field of view.”*

We have added a figure (Figure 2) showing the rotation of the PICAM boresight as below.

- *“Page 4: ‘The possibility that these dynamic events are foreshock ions (e.g.: 7), transported along the magnetic field spiral, will be investigated in a future study’ Can the authors justify this interpretation? Can foreshock ions reach these distances, despite the relatively small bow shock of Mercury? Was the Interplanetary Magnetic Field orientation during this event consistent with this interpretation? Reference (7) performed a global hybrid simulation of the ion foreshock relatively close to the bow shock, compared with the distances where the intermittent signature present in Figure 2A is observed. Can the authors rule out an instrumental component in this observation?”*

We thank the Reviewer for these suggestions. Nevertheless, concerning use of other BC data, this is not possible in this preliminary paper following PI executive internal rules. These rules state data will be put together after first separate presentations from each instrument.

Indeed, the intermittent-like high-energy enhancements are observed by PICAM when the magnetic field pointed northward during a clear magnetic field rotation eastward-northward-eastward lasting from 19:00 UT to 20:30 UT. We speculated if this rotation could be attributed to the passage of an interplanetary magnetic field structure, producing high-energy particle acceleration with respect to the background solar wind (peaking at about 600 eV). Two likely candidates have been explored: a magnetic cloud or a magnetic flux rope. The former is usually associated with a coronal mass ejection (CME), while the latter is a small, transient event. Since there were no CMEs erupting from the Sun in the previous day(s), the most probable candidate remains a flux rope. Actually, we are investigating with more details this observation, together with not yet validated magnetic field data, whose details and findings will be reported in a forthcoming paper. The sentence introduced in the revised version is the following: (line 134-143)

“Actually, the source of these intermittent structures could be the bow-shock, since their appearance is clearly associated with PICAM’s FoV pointing towards the bow shock, as opposed to the solar wind direction. Such an hypothesis is no more applicable, since a combined analysis with magnetic field preliminary data from BepiColombo/MAG (not yet validated by MAG team, not shown here), suggests that these high-energy particles are probably associated with the passage of an interplanetary magnetic flux rope. Our findings have been also validated by means of Solar Orbiter observations at a larger distance (0.6 AU). The results of our analysis will be reported in a forthcoming paper.”

- *“Page 4: ‘A more detailed analysis of these features will be performed when other BC data, such as magnetic field measurements will be available’ In my opinion, magnetic field data is necessary to provide information regarding Bepi-Colombo connectivity to the bow shock when detecting the high energy intermittent signal.”*

We are aware that the absence of other data sets and especially the MAG data implies some raw interpretation of what SERENA detects. Unfortunately, BC internal rules

state that the preliminary results should be presented separately, before joining efforts in a second phase. Nevertheless, we believe that the data we collected are of great value showing by themselves peculiar aspects of Mercury's ion environment never faced before. Naturally any possible interpretation should be seen as a possibility, to be confirmed in further studies.

- *“Page 4: ‘However, we must note that this signal was never observed in the extensive plasma ion measurements recorded by the MESSENGER mission, which orbited Mercury from 2011 to 2015.’ What do the authors mean by ‘this signal’ in this sentence? What is the origin of this signal? It is my understanding that foreshock ions have been observed by MESSENGER FIPS. This is reported, for instance, in Tracy, Patrick J. PhD Thesis (2016) <https://ui.adsabs.harvard.edu/abs/2016PhDT.....116T> and was also presented at AGU fall meetings <https://ui.adsabs.harvard.edu/abs/2018AGUFMSM51B..02G>”*

We agree that foreshock ions have been observed by MESSENGER FIPS but since our initial conjecture was that the high-energy intermittent-like signal was produced by the foreshock, FIPS was not able to sample the solar wind far away from Mercury (20-25 Rm as in the present paper). However, since the main source has been identified as a magnetic flux rope, we then removed this sentence (line 160)

- *“Page 4: ‘Just behind Mercury’s bow shock, PICAM performed ion observations within an extended energy range, including lower energies. The solar wind speed was somewhat low at about 300 km/s corresponding to ~ 550 eV (Figure 2B). Two signals at even lower energies (the bands at 38 eV and 60 eV) were clearly observed, with a variable density on time scales of 30 minutes, with sunward and anti-sunward directions, respectively. Further investigation is needed to clarify whether this signal is originating from Mercury’s interaction with the solar wind or if they are induced by spacecraft outgassing (9, 10).’ Could this signature be instrumental? I think this is important since this does not seem to be a weak signal. In particular, the number of counts is much higher than that of the solar wind, at least between ~23:49 and 00:05 UT (Figure 2B).”*

The reviewer’s comment is very useful, so that we have now added in the text (as reported here below) some more speculations about the meaning of this signal . Its real intensity respect to solar wind cannot be estimated up to the time we will show real fluxes. The showed signal is just count rates and does not allow any detailed estimate.

(line 159-169)

“The simultaneous observation by two separate BC instruments of such a low energy signal excludes the possibility that it could come from instrumental effects. The persistent presence of outgassing material around spacecraft was discovered several years ago in the surrounding of Rosetta spacecraft (11). In that case a neutral gas cloud was actually discovered and the reason why such outgassing material was staying around the spacecraft without evaporating in space suggested the idea of a sort of potential barrier causing recurrent ionization and neutralization processes. The very fact that in the case of BC such an ion potential is actually determining the existence of two distinct signals will certainly need more investigations, so that many other cruise campaigns have been planned to see when and in which conditions such a phenomenon is actually observed.”

- *“Page 6 ‘Just after this high density and hot signal, an abrupt change of plasma conditions seems to indicate the magnetopause crossing.’ What is the distance between the observed and the expected magnetopause location based on (6)?”*

The interpretation of this hot population has been revised in this version. In fact a better analysis shows that this is more likely a LLBL. In this case, the expected magnetopause is about the observed transition from magnetosheath toward LLBL. This population can be identified as low latitude boundary layer (LLBL) similarly of what has been observed in the Earth magnetosphere, marking the transition between magnetosheath and magnetosphere.

Just after this high density and hot signal at around 23:25 UT, the ion density decreased abruptly, possibly indicating that BepiColombo was inside the magnetosphere. (line 216-220 ; line 262-271)

- *“Page 6 ‘This PICAM background noise decrease was observed also during the second Venus flyby and it was interpreted as the shielding of galactic cosmic rays induced by the planet.’ Please add a reference.”*

Actually, this decrease was observed in the PICAM data, but not reported yet in a paper. The figure below shows how the decrease was evident in the PICAM data during Venus flyby#2.

The top panel shows the sky blockage of Bepicolombo during its second flyby of Venus, indicating the ratio between the blocked portion of the sky view to the whole sky. As Bepicolombo got closer to the planet, more of the spacecraft view got blocked by the planet. This blockage reached its maximum of ~37% at its closest approach at 13:51 UTC on 10 Aug 2021.

The second panel shows the background ions counts by SERENA-PICAM in the blue trace line, and the third panel shows PICAM data as energy spectrogram. In both of the latter panels it is evident that the number of background counts dropped down when the spacecraft was at its closest distance from the planet. This drop in the background ion counts of PICAM is in good correlation with the estimation from the sky blockage ratio, which is over-plotted in orange dashed line in the second plot. The increase in the ion counts between 13:51 to 14:00 UTC (jump in the blue trace line in the second panel, and the green/red pixels in third panel) are the planetary plasma ions detected on top of the background counts.

- *“Page 6 ‘...northward through the dawn flank plasma sheet,, both PICAM and MIPA...’ Please remove a comma.”*

Comma has been removed

- *“Page 6 ‘...just before the outbound magnetopause crossing occurred around 23:40 UT.’ I would add a vertical dotted line showing the location of the observed outbound magnetopause crossing.”*

There is a dashed line in Figure 3(B). This time instant is not covered by PICAM, so we cannot add the same line on panel (A) of Figure 3.

- *“Page 8: ‘... they will reveal important insight into many unknown aspects many unknown aspects of a magnetosphere deep inside the inner heliosphere, like the case of Mercury...’ Many unknown aspects is repeated, please correct.”*

We have corrected the misprint

Reply to Reviewer #3

General comment

- *“This manuscript provided first observations of Mercury's *inner magnetosphere made by the SERENA instrument package onboard the Bepi-Colombo spacecraft. This is also the first observations of Mercury's southern part of the magnetosphere, which was previously un-observed by the MESSENGER mission and not studied extensively by earlier studies. Hence this manuscript provided a first look at the possibilities results from the Bepi-Colombo mission can offer, and this work is of significance to future Mercury science. This manuscript is good as it is. The methodology, analysis and interpretation of the ion measurements of the first Mercury flyby by PICAM and MIPA is straightforward and sound. I have no major comments regarding the results presented in this manuscript.”*

We are very glad to notice that this reviewer is in favour of publishing this paper with only a minor comment which we fully understand and approve, and consequently consider in the revision.

Point-to-point responses

- *“To help the reader better understand Figure 3, I would highly recommend the authors labelled in the figure itself the different regions of Mercury's magnetosphere that each lines in Figure 3b represents. It can be confusing to have to constantly refer to the figure caption to understand which magnetospheric region did Bepi-Colombo observed. Alternatively, the use of different colors should also be considered.”*

We thank the reviewer and we added labels in Figure 3 (Figure 4 in the revised manuscript).

REVIEWER COMMENTS

Reviewer #1 (Remarks to the Author):

The authors have addressed all issues raised in the original review of the manuscript and publication can now proceed.

Reviewer #2 (Remarks to the Author):

I thank the authors for addressing my comments. As stated in my previous review report, this paper presents the first observations of Mercury's inner southern magnetosphere and nearby regions, as measured by Bepi-Colombo sensors SERENA-PICAM and –MIPA, during the first Mercury flyby on October 1, 2021. The manuscript also leaves the validation of some suggestions and conclusions for future papers. This is the case in the interpretation of high-energy particles (Figure 3A) in terms of an interplanetary flux rope and the interpretation of the two beam-like signals at about 38 eV and 60 eV (Figure 3B) in terms of spacecraft outgassing or Mercury's interaction with the solar wind. Despite this limitation, the manuscript provides measurements in a region that was not sampled by the MESSENGER mission and shows the importance these instruments have for the Bepi-Colombo mission and future investigation of the Hermean magnetosphere. I have only a few minor comments for the authors' consideration.

Lines 141-142: The authors have changed the interpretation of the intermittent high-energy observations presented in Figure 3A. The current version of the manuscript suggests these high-energy particles are probably associated with the passage of an interplanetary magnetic field flux rope. It is also stated that 'Our findings have been also validated by means of Solar Orbiter observations at a larger distance (0.6 AU)'. I think the manuscript would benefit from some additional information about this validation and its implications for Mercury's environment (~0.31 – ~0.47 au).

Lines 146-149: 'The absence of any intermittent keV-fluxes as in panel 3C could be due to the flight trajectory: if in the dusk sector the spacecraft is magnetically connected with the bow shock (the foreshock region), the contrary would happen in the opposite dawn region.' This seems to be associated with the interpretation provided in the previous version of this manuscript, where the high energy was related to bow shock/foreshock phenomena. I think this sentence needs some additional context or should be removed.

Line 157: Please define MPPE

Review of 'First observations of Mercury's inner southern magnetosphere by BepiColombo/SERENA ion sensors', by S. Orsini, et al.

Reply to Reviewer #1

General comment

- "The authors have addressed all issues raised in the original review of the manuscript and publication can now proceed."

We gratefully thank Reviewer #1 for approving publication of this paper.

Reply to Reviewer #2

General comment

- "I thank the authors for addressing my comments. As stated in my previous review report, this paper presents the first observations of Mercury's inner southern magnetosphere and nearby regions, as measured by Bepi-Colombo sensors SERENA-PICAM and -MIPA, during the first Mercury flyby on October 1, 2021. The manuscript also leaves the validation of some suggestions and conclusions for future papers. This is the case in the interpretation of high-energy particles (Figure 3A) in terms of an interplanetary flux rope and the interpretation of the two beam-like signals at about 38 eV and 60 eV (Figure 3B) in terms of spacecraft outgassing or Mercury's interaction with the solar wind. Despite this limitation, the manuscript provides measurements in a region that was not sampled by the MESSENGER mission and shows the importance these instruments have for the Bepi-Colombo mission and future investigation of the Hermean magnetosphere. I have only a few minor comments for the authors' consideration."

We thank the Reviewer #2 for the previous and present comments which have made the paper much more clear and reliable.

Point-to-point responses

- "Lines 141-142: The authors have changed the interpretation of the intermittent high-energy observations presented in Figure 3A. The current version of the manuscript suggests these high-energy particles are probably associated with the passage of an interplanetary magnetic field flux rope. It is also stated that 'Our findings have been also validated by means of Solar Orbiter observations at a larger distance (0.6 AU)'. I think the manuscript would benefit from some additional information about this validation and its implications for Mercury's environment (~0.31 – ~0.47 au)"

We do agree with the Reviewer suggestion. Hence, we have added some more comments concerning this detection (lines 141-164).

"Indeed, MAG observed the typical signature of this structure, i.e., an increase in the average magnetic field magnitude (with respect to the main background field), a decrease in the variance of magnetic field fluctuations, and a smooth rotation in one of the field components. Our preliminary findings, based on both the minimum variance analysis and the Lundquist force-free model, have been also validated by means of Solar Orbiter (SoIO) observations. Indeed, SoIO was located at a distance of 0.64 AU from the Sun (0.26 AU ahead BepiColombo) and the two spacecraft were quite well radially aligned, longitudinally separated by less than 10°, and lying on the same side of the heliospheric current sheet. Surprisingly, a similar magnetic field behaviour, in terms of average field and its fluctuations, was observed also by SoIO/MAG instrument, thus strongly corroborating the interplanetary flux rope hypothesis.

The detected flux rope expanded its radius of an order of magnitude from BepiColombo to SoLO locations, with a corresponding decrease of its main core field. Details of the results of our analysis will be reported in a forthcoming paper, as soon as the magnetic field data will be confirmed and officially declared as reliable. The actual effect over the Mercury environment would have been the subject of an interesting study, but unfortunately the solar wind structure vanished well before the flyby, and any possible internal effect was not observed. It likely produces enhanced flux transfer events and magnetic reconnection sites, together with small substorm-like activity in the nightside of the Hermean magnetosphere. However, as we will shown in the next section the Mercury's magnetosphere was in quiet conditions, thus suggesting that it reconfigured after the passage of the flux rope. Such kind of events will be further investigated during the nominal mission (after satellite orbital insertions, in late 2025), when MIO will observe the solar wind and simultaneously MPO will detect any internal reaction".

- *"Lines 146-149: 'The absence of any intermittent keV-fluxes as in panel 3C could be due to the flight trajectory: if in the dusk sector the spacecraft is magnetically connected with the bow shock (the foreshock region), the contrary would happen in the opposite dawn region.' This seems to be associated with the interpretation provided in the previous version of this manuscript, where the high energy was related to bow shock/foreshock phenomena. I think this sentence needs some additional context or should be removed."*

The whole sentence has been removed. The Reviewer is right. It came from previous interpretation.

- *"Line 157: Please define MPPE"*

MPPE is the Mercury Plasma Particle Experiment, we have indicated the extended name, and rephrased the sentences related to MPPE, just to better clarify the way to proceed for making further investigations (Editor's suggestion, lines 172-175)

"Further investigation is needed by cumulating more events statistically significant with different environmental conditions and satellite orientations, in order to clarify whether this signal is originating from Mercury's interaction with the solar wind or alternatively it is induced by spacecraft outgassing (9, 10). The Mass Spectrum Analyzer (MSA), a unit of the MPPE (Mercury Plasma Particle Experiment) consortium onboard BC-MIO, confirms the existence of a distinct double-band feature at low energies and that O⁺ is the dominant ion species (Lina Hadid, private communication)."

REVIEWER COMMENTS

Reviewer #2 (Remarks to the Author):

I thank the authors for addressing all my previous comments.